# The Efficacy and Cost-Effectiveness of Replacing Whole Apples with Sliced in the National School Lunch Program

**DOI:** 10.3390/ijerph182413157

**Published:** 2021-12-14

**Authors:** Shelly Palmer, Jessica Jarick Metcalfe, Brenna Ellison, Toni Kay Wright, Lindsey Sadler, Katherine Hinojosa, Jennifer McCaffrey, Melissa Pflugh Prescott

**Affiliations:** 1Department of Food Science and Human Nutrition, University of Illinois at Urbana Champaign, Urbana, IL 61801, USA; smpalmer@illinois.edu (S.P.); jarick2@illinois.edu (J.J.M.); klh2@illinois.edu (K.H.); 2Department of Agricultural Economics, Purdue University, West Layfayette, IN 47907, USA; bdelliso@purdue.edu; 3Office of Extension and Outreach, University of Illinois at Urbana Champaign, Urbana, IL 61801, USA; tkwright@illinois.edu (T.K.W.); lrhicks1987@gmail.com (L.S.); jmccaffr@illinois.edu (J.M.)

**Keywords:** school nutrition, food waste, implementation science, behavioral economics

## Abstract

The National School Lunch Program (NSLP) serves 29.6 million lunches each day. Schools must offer ½ a cup of fruit for each lunch tray. Much of this fruit may be wasted, leaving the schools in a dilemma. The objectives of this study were to evaluate the consumption of whole vs. sliced apples and determine the cost-effectiveness of the intervention. Researchers weighed apple waste at baseline and three post-intervention time points in one rural Midwest school. The costs of the intervention were collected from the school. The cost-effectiveness analysis estimates how often apples need to be served to offset the costs of the slicing intervention. A total of (*n* = 313) elementary student students participated. Students consumed significantly more sliced as compared to whole apples in intervention months 3 (*β* = 21.5, *p* < 0.001) and 4 (*β* = 27.7, *p* < 0.001). The intervention cost was USD 299. The value of wasted apple decreased from USD 0.26 at baseline to USD 0.23 wasted at post-intervention. The school would need to serve 9403 apples during the school year (54 times) to cover the expenses of the intervention. In conclusion, serving sliced apples may be a cost-effective way to improve fruit consumption during school lunch.

## 1. Introduction

Approximately 60% of children ages 2–4 years meet or exceed the 2020–2025 Dietary Guidelines for Americans (DGA) recommendations for fruit [1]. By late adolescence, fruit consumption patterns decline to about half of the recommended intake [1]. According to the 2015–2016 NHANES, whole fruit consumption among youth ages 6–11 years was less than 1 cup/day, [2] falling short of the 2020–2025 DGA recommendation of 1–2 cups/day [1]. Fruit is an excellent source of fiber, which is under-consumed in America and [3] worldwide [4]. Regular fruit consumption is associated with beneficial cardiovascular outcomes [5,6]. In 2017, 2 million diet-related deaths were attributed to low fruit consumption globally [4]. Additionally, wasted food is a major concern across the global supply chain, [7] as inputs to food production, such as land, waste, and energy, are squandered when food is discarded instead of eaten [8].

The National School Lunch Program (NSLP) serves approximately 29.6 million lunches each day [9]. The NSLP requires schools to offer a minimum of 1/2 a cup of fruits for grades K-8 [10,11]. Although NSLP fruit selection significantly increased after the implementation of the Healthy, Hunger-Free Kids Act (HHFKA) from 54% in 2012 to 66% in 2014, fruit consumption patterns did not significantly increase [12]. Fruits and vegetables are the largest contributors to NSLP waste [13]. For example, a school lunch costs approximately USD 1.80, and about USD 0.19 per tray was wasted on fruit in Boston middle schools in 2007–2009 [14].

Previous fruit slicing research among middle school and K-4th-grade students suggests that offering pre-sliced fruit instead of whole fruit is associated with increases in fruit consumption [15,16]. However, there is a gap in the literature on how to sustainably incorporate this evidence- based practice into NSLP. Implementation Science (IS) examines the adoption and sustainability of evidence-based innovations in clinical or public health settings [17]. IS determinants, such as costs, are contextual factors [18] that may influence the adoption of innovations in the NSLP given the budget constraints of school nutrition programs. The first objective of this study was to evaluate the efficacy of replacing whole apples with sliced apples to improve fruit consumption. The second objective was to determine the cost-effectiveness of serving sliced apples in an NSLP [19].

## 2. Materials and Methods

### 2.1. Setting and Participants

One rural elementary school providing the NSLP was recruited to participate in the study through the partnership with the University of Illinois Extension community nutrition programs. All students in the elementary school were eligible to participate in the study if they participated in the NSLP. NSLP meals for students in kindergarten through to 2nd grade had a serve policy (no choice in meal components), while 3rd–5th-grade students utilized an offer vs. serve format. Students in 3rd–5th-grade could choose between two entrees and select a fruit and/or vegetable. Students in 5th-grade could also use the salad bar. All students at the school received free school lunches. Apples were selected for the intervention since the school nutrition staff estimated that they are offered as a fruit option daily for 3–5th-grade and served 2–3 times/month for kindergarten−2nd grade. The study was approved by the University of Illinois Institutional Review Board (IRB # 19233).

### 2.2. Intervention

In the first month of data collection (baseline), whole apples were served. In the following three intervention months, the equivalent of a whole apple was served in slices, without removing the skin.

### 2.3. Data Collection for the Intervention

The study took place from February through May 2019. Passive parental consent letters were sent home to parents, and children verbally assented to participate. The researchers aimed to obtain a random sample of approximately 25% of the students participating in the NSLP each day of data collection. In order to randomly select eligible participants, participant ID labels were randomly placed on 40% of the students’ trays before lunch service on data collection days. Students who received a tray with a label were invited to participate in the study. The same menu was served for each day of data collection. The students were not informed ahead of time when the research team would be collecting data.

Four to five portions of each served meal component (entrée (which combined protein and grain), fruit, vegetable, and milk options) were weighed to determine an average reference weight before lunch was served [20]. Food selection was assessed via images of the 3rd–5th-grade students’ lunch trays as they exited the lunch lines. Since salad bar items were self-served, the items students selected from the salad bar were weighed before students sat down to eat (i.e., the salad bar food items reference weight) and after the lunch period ended. Once lunchtime was over, each participant’s uneaten portions of food were weighed and recorded to the nearest 0.5 g using a digital scale (Taylor Professional commercial food scale) [20]. The costs of purchasing the apples, equipment, and labor were collected from school nutrition staff.

### 2.4. Data Analysis

For each participant, all pre- and post-meal component weights were dual-entered by two trained research assistants using a standardized electronic form. Data were compared against the selection photos and data collection sheets by research assistants to check for accuracy and resolve any discrepancies. Meal component consumption was calculated by subtracting the reference weight from the amount of food wasted (in grams). Multiple imputation, a modern imputation method that is less biased than traditional approaches such as mean substitution or case-wise deletion, was used to impute data that were missing [21,22]. Ten imputed data sets were created for data that were missing using the Fully Conditional Specification method. These ten data sets were averaged to create a single imputed data set, and pooled estimates (aggregated imputed values) were used in all analyses [21]. Missing data rates ranged from 0 to 11% across study variables. Data were analyzed using SPSS Statistical Software (IBM, Armonk, NY, USA). Estimated marginal means for all meal components were calculated for each month. Percent meal component consumption was calculated by dividing the amount consumed by the reference weight. Multiple linear regression models were used to compare the fruit consumption outcomes of the baseline month compared to the intervention months. Fruit consumption was controlled for gender, grade, and an index variable, which included the consumption of non-fruit meal components. Total meal consumption analyses controlled for gender and grade. Apple variety was also controlled for in the 3rd–5th-grade sampled population. For the baseline month (when apples were whole, not sliced), the average weight of the inedible portion of the apple core was subtracted to estimate the weight of the edible apple slices. Significance for all analyses was set to *p* < 0.05.

## 3. Results

A total of 313 student trays were sampled across the four months. An average of 37% of the eligible students were sampled. Table 1 outlines school-level and participant-level demographics.

At baseline, 97% of the students sampled had apples on their trays. Fruit selection rates remained high during each month of the intervention; 79% in March, 93% in April, and 88% in May. Linear regression findings for the total population of total meal and fruit consumption are outlined in Table 2. At baseline, students consumed 20% of the apples served. The percent of apple consumed significantly increased compared to baseline in April (*β* = 21.5, *p* < 0.001) and May (*β* = 27.7, *p* < 0.001). The increase of 21.5 percentage points of apple consumption in April equates to a reduction of 49.6 g of apple wasted per student. The increase of 27.7 percentage points of apple consumption in May equates to a reduction of 45.3 g of apple wasted per student. Total meal consumption increased during the intervention months with the greatest consumption in May (*β* = 10.8, *p* < 0.01). 

Total meal and fruit consumption results of the 3rd–5th-grade students who selected apples are shown in Table 3. When apple variety was adjusted for, fruit consumption in April (*β* = 16.8, *p* < 0.05) and May (*β* = 17.9, *p* < 0.05) remained statistically higher when compared to the baseline month. When comparing meal component consumption across the intervention months, milk consumption significantly increased in April (*β* = 21.8, *p* < 0.05), otherwise there were no other significant findings.

The served apples cost USD 0.32 each. The apple slicer (USD 299) was the only cost incurred from the intervention. Slicing the apples took two school nutrition staff members about one hour to prepare and portion on trays. No additional labor or overtime hours were needed for slicing the apples relative to the baseline (when whole apples were served). The average percent apple waste in the baseline month was 81%, with an average of USD 0.26 of apples discarded (per apple). The average percent apple waste in the intervention months was 71%, with an average of USD 0.23 of apples discarded (per apple). On average, a difference of USD 0.0318 per apple was saved from the baseline month relative to the intervention months. To pay off the slicer expense, 9403 (USD 299/USD 0.0318) apples need to be served. When apples are served to kindergarten through to 5th graders at the target school, approximately 175 apples are distributed each time. Apples will need to be served approximately 54 times at this school to offset the cost of the slicing appliance.

## 4. Discussion

This study evaluated the efficacy of replacing whole apples with sliced apples to improve fruit consumption and determined the cost effectiveness of serving sliced apples in the NSLP. When sliced apples were served, apple consumption significantly increased in months 3 and 4, and total meal consumption increased relative to baseline in the final month. The intervention did not result in increased labor costs. The apples were not a new expense for the school, as they were already providing apples as part of the NSLP. Since the older students have the choice of selecting an apple each day and younger students are served apples 2–3 times per month, reaching the goal of serving apples 54 times throughout the year is realistic for this school. However, some schools may not be able to cover the equipment costs of a slicer and may need to seek grants or other funding to cover this expense.

Previous fruit slicing research among elementary school students found that significantly more sliced vs. whole oranges were consumed, although there was no significant difference in apple consumption due to slicing [16]. In a study of middle school students, after introducing sliced apples, the percent of students consuming more than half of their apple significantly increased by 73% [15]. In the present study, the K-2nd grade students exhibited greater post-intervention increases in consumption relative to older students. These differences may be driven by the developmental stage of the younger children, but it is also important to note that these younger grades did not experience offer vs. serve, while the older grades did. Thus, it is possible that sliced fruit may be particularly convenient for students who are served all the NSLP food components.

Schools may purchase pre-packaged, sliced fruit instead of manually slicing the fruit [23]. Purchasing pre-sliced apples can be more costly for schools, as a five-pound tray of pre-sliced apples was more than four times the cost of a whole apple [24]. Although individually packaged products may cost more, fewer staff may be needed on the service line, and pre-packaged products may facilitate moving more students through the line quicker [25]. This is important since less time spent in line facilitates more seated lunch time, which has been causally linked to decreased food waste [26]. Pre-packaged food items were allowed in all of the 24 states, which have a share table policy (to recover unwanted food that would otherwise be landfilled), whereas 20 states permitted washed fruit with an edible peel [27]. This suggests that pre-packaged, sliced apples may allow greater food recovery among schools utilizing share tables.

The findings of this study are not without limitations. Most notably, there was no control group, so it is impossible to rule out unmeasured influences on apple consumption. Convenience sampling was used to select the participating school. The results are limited to a rural school district in Illinois and may not be generalizable to urban and suburban settings. Students’ eating patterns may have changed on the days when research staff were present. The apple slicing intervention was implemented over a three-month time period, so long-term studies are needed to determine if these short-term improvements continue over time. Two different apple varieties were offered during data collection, which was not part of the research design, but apple variety was controlled for in the statistical analysis for the 3rd–5th-grade sub-sample. Further research is needed to investigate other IS determinants of slicing interventions, such as acceptability, feasibility, and appropriateness. More research is needed to understand whether there are cost or waste reduction advantages of serving manually or pre-packaged sliced apples.

## 5. Conclusions

These findings suggest that offering sliced versus whole apples may increase the amount of fruit consumed among elementary school students. The magnitude of the decrease was highest among younger students, suggesting grades K-2 may be particularly appropriate for slicing interventions. The intervention was cost effective for the school as no overtime or extra staff were needed to slice the apples, which may suggest the practicality of replicating the apple slicing intervention in other schools.

## Figures and Tables

**Table 1 ijerph-18-13157-t001:** Student-level and participant-level demographic characteristics of intervention school.

**School-Level Characteristics**	
Total Enrollment	212
Grades	Kindergarten-5th-grade
Percent Meal Participation *^,1^	63%
Predominant Race/Ethnicity	White
**Participant Characteristics**	
Total Number of Trays Sampled	313
Average Number of Trays per Month *	78
Gender	51% Female 49% Male
Grades	52% Kindergarten-2nd 15% 3rd 17% 4th 17% 5th

* Displayed as average across all months of data collection. ^1^ Data Source: Illinois State Board of Education.

**Table 2 ijerph-18-13157-t002:** Linear regressions for percent consumption of fruit and total meal by month for full sample size, *n* = 313.

	Fruit Component (Apple)			Total Meal		
	% Consumed	Beta	*p*	% Consumed	Beta	*p*
**February**	20%			46%		
**March**	24%	4.8	0.24	48%	1.6	0.65
**April**	41%	21.5	<0.001	50%	3.8	0.33
**May**	47%	27.7	<0.001	58%	10.8	0.004

*Note*. All analyses control for participant gender and grade, and fruit analyses also control for meal consumption index variable (average % consumed for other non-fruit meal components). *p*-values are associated with differences between reference month (February) and each month of the intervention (March, April, and May).

**Table 3 ijerph-18-13157-t003:** Linear regressions for percent consumption of fruit and total meal by month for 3rd–5th graders who selected apples, *n* = 115.

	Fruit Component (Apple)			Total Meal		
	% Consumed	Beta	*p*	% Consumed	Beta	*p*
**February**	24%			46%		
**March**	32%	7.7	0.33	50%	3.3	0.56
**April**	41%	16.8	0.04	52%	5.4	0.37
**May**	43%	17.9	0.04	50%	3.3	0.57

*Note*. All analyses control for participant gender, grade, and apple variety, and fruit analyses also control for the meal consumption index variable (average % consumed for other non-fruit meal components). *p*-values are associated with differences between reference month (February) and each month of the intervention (March, April, and May).

## Data Availability

The data presented in this study are available on request from the corresponding author. The data will eventually be made public via the Illinois Data Bank once related analyses are completed and published.

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
