# Peer review of "The Efficacy and Cost-Effectiveness of Replacing Whole Apples with Sliced in the National School Lunch Program"

_ijerph, 2021, doi:10.3390/ijerph182413157_

Round 1
Reviewer 1 Report
Introduction, it could be important to mention, in scientific terms, possible health benefits in the increased consumption of fruit and green products.
Methods, suggested clarifying the criteria for inclusion in the study, the randomization process, as well as the description of the metric processes.
the statistical n/sample is even very small. Could it be increased?
Results, the results presentation tables are not very appealing / not very suggestive. Could be improved.
Discussion, I would like to see the concept of "costeffectiveness" more substantiated, comparing it with other studies. Could in the discussion deepen the idea of the environmental impact of the measure, in the context of creating a "green" economy
It does not discuss whether there is any impact on the nutritional value of the apple... taking the skin off can have some problems!
Conclusion, I suggested only responding to the proposed outcomes and nothing more. All other comments can be made in the discussion.
Author Response
Thank you to the reviewer for their time and feedback. We appreciate this opportunity to strengthen our manuscript. Our response to the reviewer's feedback are in italicized below. Please see the attached revised manuscript. Track changes are included in the word document.
Introduction, it could be important to mention, in scientific terms, possible health benefits in the increased consumption of fruit and green products.
The introduction was edited to include more information about the health benefits of fruit consumption a well as the negative impact of wasted food to the environment. See lines 32-34.
Methods, suggested clarifying the criteria for inclusion in the study, the randomization process, as well as the description of the metric processes.
A sentence was added in line 58 to describe the inclusion criteria for the study. The intervention was not randomly allocated. However, the sample was randomly selected from the eligible participants. Edits were made to lines 76-78 to make this process clearer. Regarding the metric processes, lines 91-92 describe how waste was measured with a digital scale to the nearest 0.5 grams.
The statistical n/sample is even very small. Could it be increased?
Yes, we agree with the reviewer that this is a small sample size. Given that the data are already collected, it is not possible to increase the sample size. We conducted multiple imputations to address missing data in the interest of having the largest possible sample size. Please see lines 91-97.
Results, the results presentation tables are not very appealing / not very suggestive. Could be improved.
We are not sure what this reviewer is suggesting to change about these tables. We are open to editing the way results are presented but are unsure as to how the reviewer would prefer us to move forward. After considering the reviewer’s comment, we noticed that there was some repetition between the text and Table 1. The repetitive text has been omitted.
Discussion, I would like to see the concept of "costeffectiveness" more substantiated, comparing it with other studies. Could in the discussion deepen the idea of the environmental impact of the measure, in the context of creating a "green" economy
The economic implications of high fruit waste in school lunch settings are mentioned in the introduction: “The NSLP requires schools to offer a minimum of ½ cup of fruits for grades K-8. Although NSLP fruit selection significantly increased after the implementation of the Healthy, Hunger-Free Kids Act (HHFKA) from 54% in 2012 to 66% in 2014, fruit consumption patterns did not significantly increase. Fruits and vegetables are the largest contributors to NSLP waste. For example, a school lunch costs approximately $1.80, and about $0.19 per tray was wasted on fruit in Boston middle schools in 2007-2009.”
The environmental implications of pre-packaged pre-sliced fruit in school lunch settings are mentioned in the discussion: “Schools may purchase pre-packaged, sliced fruit instead of manually slicing the fruit. Purchasing pre-sliced apples can be more costly for schools, as a five-pound tray of pre-sliced apples was more than four times the cost of a whole apple. Although individually packaged products may cost more, fewer staff may be needed on the service line, and pre-packaged products may facilitate moving more students through line quicker. This is important since less time spent in line facilitates more seated lunch time, which has been causally linked to decreased food waste. Pre-packaged food items were allowed in all of the 24 states which have a share table policy (to recover unwanted food that would otherwise be landfilled), whereas 20 states permitted washed fruit with an edible peel. This suggests that pre-packaged, sliced apples may allow greater food recovery among schools utilizing share tables.”
It does not discuss whether there is any impact on the nutritional value of the apple... taking the skin off can have some problems!
The skin was not removed when the apples were sliced. The following sentence was added to the methods section: “In the following three intervention months, the equivalent of a whole apple was served in slices, without removing the skin.” See line 72.
Conclusion, I suggested only responding to the proposed outcomes and nothing more. All other comments can be made in the discussion.
Edited as suggested. The two sentences that were not related to the proposed outcomes were moved to the discussion and are now located in lines 233-237.

Reviewer 2 Report
It's an interesting topic. Here are some comments:
1.The "fruit" in Table 3 should be "apple"
2.Are there any other promotions to encourage students to consume more fruit? Because the growth in total fruit consumption was only higher than the growth in apple consumption. Other factors, such as seasonal changes, also affect fruit consumption. Was the increase in apple intake influenced by other interventions than the "slice" intervention? If so, it should be described in the restrictions section.
3.Detailed intervention progress should be described in the methods section.
4.In abstract terms, "the intervention cost was $299, of which $0.26 apples were wasted at baseline and $0.23 apples were wasted after the intervention. It should be improved.
5.In the current study, the percentage of fruit consumption was only given over four months. In general, fruit is consumed more in the warm season than in the cold season. Is the percentage of fruit consumption likely to decline over time? If this happens, the results of the cost-benefit analysis of current research will be challenged.
6.The text describes a cost-benefit analysis. Is it possible to provide a graphical description form to make it clearer?
Author Response
Thank you to the reviewer for their time and feedback. We appreciate this opportunity to strengthen our manuscript. Our response to each reviewer comment is provided in italics below. Please see the edited manuscript, attached. Track changes are included in the word document.
It's an interesting topic. Here are some comments:
1.The "fruit" in Table 3 should be "apple"
Thank you for the comment on clarification. Our original submission used the term fruit as fruit is one of the 5 meal components of the National School Lunch Program, but we understand how this could be confusing. In response to the reviewer’s comment, we edited the column heading to say “fruit component (Apple)” in tables 2 and 3 to show the distinction between meal component consumption vs. total meal consumption.
2.Are there any other promotions to encourage students to consume more fruit? Because the growth in total fruit consumption was only higher than the growth in apple consumption. Other factors, such as seasonal changes, also affect fruit consumption. Was the increase in apple intake influenced by other interventions than the "slice" intervention? If so, it should be described in the restrictions section.
Offering sliced vs. whole apples was the only fruit-promoting strategy during the 4 months of data collection during the spring semester. The footnote addition mentioned above may assist with the understanding that apples were the only fruit served and analyzed during the 4 months of data collection. Also, we are transparent the limitation of not being able to rule out unmeasured influences on apple consumption in lines 209-210 of the discussion: “The findings of this study are not without limitations. Most notably, there was no control group so it is impossible to rule out unmeasured influences on apple consumption.”
3.Detailed intervention progress should be described in the methods section.
To address the reviewer’s comment, we have added a subheading so that the intervention is clearly identified. See lines X.
4.In abstract terms, "the intervention cost was $299, of which $0.26 apples were wasted at baseline and $0.23 apples were wasted after the intervention. It should be improved.
Thank you for the comment on clarification. This sentence in the abstract has been edited to state the following: “The intervention cost $299. The value of wasted apple decreased from $0.26 at baseline to $0.23 wasted at post-intervention.”
5.In the current study, the percentage of fruit consumption was only given over four months. In general, fruit is consumed more in the warm season than in the cold season. Is the percentage of fruit consumption likely to decline over time? If this happens, the results of the cost-benefit analysis of current research will be challenged.
The reviewer mentions fruit consumption increasing during warmer seasons. If true, that would mean the reviewer would expect fruit consumption to increase over time since baseline of the current study was in February (Winter) and ended in May (Spring). While the reviewer’s point about fruit consumption may reflect general consumption trends, evidence from our research suggests that this is not the case within school lunch programs. This is a unique setting where fruit is required to be offered daily with few other food choices to compete with the fruit. School food consumption, including fruit, often decreases as the school year progresses. For example, our same research team collected data during the same academic year as the present study in other schools in the same state. In these 3 schools, fruit consumption typically peaked during the intervention and then declined during the March-May period. (See figure 2 in doi:10.3390/ijerph17113971). Thus, to answer the reviewer’s question, we feel it is unlikely that fruit consumption would have improved during the March-May period given that other schools studied in the same state during the same period had decreased fruit consumption over the course of the school year.
6.The text describes a cost-benefit analysis. Is it possible to provide a graphical description form to make it clearer?
As a point of clarification, this study conducts a cost-effectiveness analysis rather than a cost-benefit analysis. We feel that the edited version of the cost-effectiveness analysis in the abstract is sufficient to add clarity to these findings.
